# Acute Hepatitis B and Unusual Follow-Up in a 16-Year-Old Boy: Case Report

Carlo Bieńkowski [1,2,3,*], Monika Kowalczyk [4], Magdalena Pluta [2,3] and Maria Pokorska-Śpiewak [2,3]

1    Doctoral School, Medical University of Warsaw, Żwirki i Wigury 61, 02-091 Warsaw, Poland
2    Department of Children's Infectious Diseases, Medical University of Warsaw, Wolska 37, 02-091 Warsaw, Poland; magdalena.pluta@lekarz.eu (M.P.); mpspiewak@gmail.com (M.P.-Ś.)
3    Hospital of Infectious Diseases, 01-201 Warsaw, Poland
4    Student's Scientific Group, Department of Children's Infectious Diseases, Medical University of Warsaw, Wolska 37, 02-091 Warsaw, Poland; mkkowalczyk1@gmail.com
*    Correspondence: carlo.bienkowski@gmail.com; Tel.: +48-22-3355-301

**Abstract:** In this case report, we present a 16-year-old Ukrainian boy with acute hepatitis B. He had not been previously vaccinated against hepatitis B. Possible sources of infection included: a tattoo made at home, a finger cut made with hairdresser scissors during work, and unprotected sexual encounters. The clinical course of the disease was typical with jaundice and elevated amino-transferases levels without liver failure. During the follow-up visit 16 months after the onset of the disease, chronic hepatitis b was excluded but an ulcer around his anus was found. Additional tests for sexually transmitted diseases were ordered and they were positive for syphilis. The extended interview revealed that the patient had several unprotected bisexual encounters, which may have indicated a potential source of infections including the hepatitis B virus (HBV). The reported case shows that despite the significant decrease in the hepatitis B prevalence in Poland, the infection is still possible. It is necessary to conduct epidemiological interviews regarding sexually transmitted diseases in teenagers, especially when a blood-borne disease has been diagnosed.

**Keywords:** hepatitis B; syphilis; vaccinations; infections; blood-borne pathogens

## 1. Introduction

Hepatitis B is caused by Hepatitis B virus (HBV), which can cause acute or chronic infection. The infection occurs through contact with infected blood, sexual contact with infected people, and non-sterile sharp medical equipment (needles and surgical instruments) contaminated with the blood of an infected person or by sharing the equipment while using intravenous drugs [1].

Hepatitis B symptoms appear in half of HBV-infected patients. Characteristic symptoms include malaise, lack of appetite, jaundice, dark urine, and digestive disorders [2]. Chronic hepatitis is an infection lasting over 6 months. After several years, it may lead to the development of liver cirrhosis. A chronically infected person is also at risk for hepatocellular carcinoma [3].

After the introduction of routine vaccination against hepatitis B for all infants in Poland (1994–1996), the incidence of hepatitis B decreased significantly and nowadays cases among children are very rare [4].

In Ukraine, which is bordering Poland, a study by Khetsuriani et al. on the seroprevalence of hepatitis B virus infection markers among children in Ukraine revealed that HBV seroprevalence in this population is low [5].

The aim of this case report was to highlight the role of vaccinations in preventing infections, as well as the importance of interviews regarding sexually transmitted diseases in teenagers presenting with blood-borne diseases.

## 2. Case Report

A 16-year-old Ukrainian boy was admitted to the Department of Children's Infectious Diseases in Warsaw, Poland, due to jaundice lasting for four days with malaise, diarrhoea, and vomiting. The boy has lived in Poland for 1.5 years; he was working (training) in the profession of a hairdresser. The medical interview revealed that he had a tattoo made at home by his mother's partner six months earlier and a finger was also cut by hairdressing scissors six weeks before the admission. The patient confirmed that he had two secured heterosexual encounters during the last six months. In Ukraine, he had been vaccinated against tuberculosis, diphtheria, tetanus, pertussis, poliomyelitis, measles, mumps, and rubella; he has not been vaccinated against hepatitis B. He negated any dietary errors and had never had blood transfusion nor surgery. He was once hospitalized due to acute gastroenteritis (in Ukraine). The physical examination revealed jaundice, hepatomegaly (2 cm below the rib arch), and a tattoo on the right forearm. Laboratory tests showed a significantly elevated level of aminotransferases (alanine aminotransferase (ALT) 2439 IU/L and aspartate aminotransferase (AST) 1418 IU/L) (Table 1). Serological testing was positive for HBsAg and anti-Hbc IgM antibodies, and negative for hepatitis A and hepatitis C viruses. The HBV viral load was 5.82 copies/mL and HBV genotype A was confirmed. Acute hepatitis B was diagnosed and the patient was treated conservatively. There were no signs of hepatic failure. During observation, hepatic parameters were firstly elevated, but after 14 days they improved and the patient was discharged home with the recommendation to appear for the follow-up examinations after six months (in order to exclude or confirm chronic hepatitis B). The patient did not appear for the visit. After 15 months, he wrote a message on Facebook to his practitioner asking if he had to report on a follow-up, as he felt good. A visit in the clinic was arranged for him. Hepatic parameters were normal, testing towards the hepatitis B antigen was negative, and the chronic HBV infection was excluded. The extended interview revealed a urinary tract infection a year before and unprotected sexual encounters of both homo and heterosexual relations. During physical examination, a small ulcer around his anus was found. Thus, additional tests for HIV, syphilis, gonorrhoea, and chlamydia trachomatis were ordered and they were all negative except for syphilis (Table 1). Due to risky sexual behaviour, the patient was offered pre-exposure prophylaxis. The patient applied for further treatment of syphilis to the Clinic of Dermatology and Venereology in Warsaw.

**Table 1.** Laboratory features of the patient on admission 7 days after admission, 14 days after admission, and 16 months after discharge of the patient.

| Laboratory Test | On Admission | 7 Days | 14 Days | 16 Months |
|---|---|---|---|---|
| ALT [10–70 IU/L]* | 2439 | 2818 | 661 | 41 |
| AST [10–59 IU/L]* | 1418 | 1427 | 125 | 32 |
| GGTP [15–73 IU/L]* | 110 | 99 | 42.8 | 8.7 |
| Total bilirubin [3–22 μmol/L]* | 131 | 157 | N/A | N/A |
| CRP [<10 mg/L]* | 13 | N/A | N/A | N/A |
| INR [0.77–1.42]* | 1.3 | 1.29 | 1.16 | 1.2 |
| HbSAg | + | N/A | N/A | - |
| Anti-Hbc IgM | + | N/A | N/A | + |
| Anti-HIV | N/A | N/A | N/A | - |
| Anti-HAV | - | N/A | N/A | + |

**Table 1.** *Cont.*

| Laboratory Test | On Admission | 7 Days | 14 Days | 16 Months |
|---|---|---|---|---|
| Anti-HCV | - | N/A | N/A | - |
| VDRL | N/A | N/A | N/A | 1:64 |
| FTA Abs | N/A | N/A | N/A | + |
| CT/NG | N/A | N/A | N/A | - |

[]* Normal values. Abbreviations: ALT, alanine aminotransferase; AST, aspartate aminotransferase; GGTP, gamma glutamyl transpeptidase; CRP, C-reactive protein; INR, international normalized ratio; HbSAg, hepatitis B surface antigen; Hbc, hepatitis B core; HAV, hepatitis A virus; HCV, hepatitis C virus; VDRL, venereal diseases research laboratory; FTA Abs, fluorescent treponemal antibodies absorption test; CT/NG, chlamydia trachomatis/Neisseria gonorrhoeae; and N/A, not available.

### 3. Discussion

Routine vaccinations against HBV of all newborns in Poland were introduced in 1994–1996, which contributed to a significant decrease in the incidence of hepatitis B (from 34.5/100,000 in 1993 to 4.11/100,000 in 2004) [4]. According to Polish law, every person staying on the territory of the country for more than 3 months is covered by the national vaccination program [6]. The patient has lived in Poland for 1.5 years, yet he has not been vaccinated against hepatitis B. Unvaccinated immigrants pose an epidemiological threat. Moreover, the fact that the unvaccinated boy was infected with HBV in Poland indicates that the virus is still circulating in our population, despite the improvement in the epidemiological situation. This shows the importance of making both physicians and immigrants aware of the possibility of being vaccinated [7].

In the future, social media can be used as a source of a direct contact with a practitioner. Probably, if the boy had not contacted his physician, there would have been no follow-up and he would have not been ruled out of having a chronic infection. Moreover, the patient would not have been diagnosed with syphilis. Untreated syphilis can lead to numerous and serious complications, and this was prevented by quick contact with the physician via social media [8].

It is crucial to deepen the medical interview with an adolescent in regards to sexually transmitted infections (STIs), especially when blood-borne infections are diagnosed. When one sexually transmitted disease is diagnosed, the patient should also be tested for other STIs such as hepatitis C, human immunodeficiency virus, chlamydia trachomatis, Neisseria gonorrhoeae, and syphilis infections [9].

Pre-exposure prophylaxis (PrEP) has significantly reduced the risk of HIV acquisition. PrEP is an administration of antiretroviral drugs in HIV-negative patients in order to prevent the infection. It is recommended in patients with a history of risky sexual behaviour. However, administration of antiretroviral drugs may bring the illusory sense of safety during sexual encounters and may be the cause of an increased risk of acquiring STIs other than HIV [10].

In conclusion, HBV infection may be an epidemiological problem among non-vaccinated people. Therefore, routine vaccination against HBV should be carried on among children. After a confirmed STI diagnosis in teenagers, testing and deepening the interview regarding other sexually transmitted diseases is necessary. Moreover, PrEP administration in adolescents with a history of risky sexual behaviour should be taken into consideration.

**Author Contributions:** Conceptualization, C.B., M.K., M.P. and M.P.-Ś.; methodology, C.B. and M.P.-Ś.; software, C.B. and M.K.; validation, C.B. and M.P.-Ś.; formal analysis, C.B. and M.P.-Ś.; investigation, C.B., M.K., M.P. and M.P.-Ś.; resources, C.B.; data curation, C.B.; writing-original draft preparation, C.B., M.K., M.P. and M.P.-Ś.; writing review and editing, C.B. and M.P.-Ś.; visualization, C.B.; supervision, M.P.-Ś.; project administration, C.B. and M.P.-Ś.; funding acquisition: C.B. All authors have read and agreed to the published version of the manuscript.

**Funding:** Article processing charges were funded by Foundation for Science Development in Hospital of Infectious Diseases in Warsaw, Wolska 37, 01-201 Poland.

**Institutional Review Board Statement:** Not applicable.

**Informed Consent Statement:** Informed consent statement was obtained from the patient.

**Data Availability Statement:** Data sharing not applicable.

**Conflicts of Interest:** The authors declare no conflict of interest.

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
