# Peer review of "Acute Hepatitis B and Unusual Follow-Up in a 16-Year-Old Boy: Case Report"

_pediatrrep, doi:10.3390/pediatric13030062_

Round 1

Reviewer 1 Report

It would be interesting to explain in more detail the population origin of the adolescent as well as the possible public health measures to be implemented so that these cases are not repeated in young people.

Author Response

Thank you for this remark. It certainly helped us improve our case report. Therefore we have it implemented. 

Reviewer 2 Report

An interesting case, well described and presented.
There is no high scientific contribution, but it could be used for future reference and as a clinical experience.

Author Response

Thank you for the acknowledgement of our work.

Reviewer 3 Report

This is a case report describing the Acute Hepatitis B and unusual follow-up in a 16-year old boy.
This was an engaging article for me to read. While reading the manuscript, I found no mistakes or major issues from a scientific point of view. The authors properly addressed the main issue.

Remarks are as follows:

1) Line 20: towards > for
2) Line 47: from > for
3) Line 63: parameters firstly elevated > parameters were firstly elevated
4) Please revise grammar in the phrase at lines 73-74.
5) Line 73: exept > except
6) Line 86: This shows how important > This shows the importance
7) Line 103: security > safety
8) Please rephrase the sentence at line 105. The current one seems too obvious.
9) Line 108: history > a history
10) Line 110: admittion > admission
11) Line 125: not applicable > not available

Author Response

Thank you for all your comments. They are all very valuable and helped us improve our case report. Therefore all of them are implemented.

This manuscript is a resubmission of an earlier submission. The following is a list of the peer review reports and author responses from that submission.